# Photoactive Pore Matrix for In Situ Delivery of a Photosensitizer in Vascular Smooth Muscle Cells Selective PDT

**DOI:** 10.3390/ma12244110

**Published:** 2019-12-09

**Authors:** Magdalena Wawrzyńska, Maciej Duda, Iwona Hołowacz, Aleksandra Kaczorowska, Agnieszka Ulatowska-Jarża, Igor Buzalewicz, Wojciech Kałas, Edyta Wysokińska, Dariusz Biały, Halina Podbielska, Marta Kopaczyńska

**Affiliations:** 1Department of Emergency Medical Service, Wroclaw Medical University, Parkowa 34, 51-616 Wroclaw, Poland; mag.wawrzynska@gmail.com; 2Department of Biomedical Engineering, Wroclaw University of Science and Technology, Wybrzeze Wyspianskiego 27, 50-370 Wroclaw, Poland; maciej.duda@pwr.edu.pl (M.D.); iwona.holowacz@pwr.edu.pl (I.H.); aleksandra.kaczorowska@pwr.edu.pl (A.K.); agnieszka.ulatowska-jarza@pwr.edu.pl (A.U.-J.); igor.buzalewicz@pwr.edu.pl (I.B.); halina.podbielska@pwr.edu.pl (H.P.); 3Department of Experimental Oncology, Ludwik Hirszfeld Institute of Immunology and Experimental Therapy, Polish Academy of Sciences, ul. Rudolfa Weigla 12, 53-114 Wroclaw, Poland; wojciech.kalas@hirszfeld.pl (W.K.);; 4Department and Clinic of Cardiology, Wroclaw Medical University, Borowska 213, 50-556 Wrocław, Poland; dariusz.bialy@umed.wroc.pl

**Keywords:** photoactive matrix, intravascular photodynamic therapy, cardiovascular stent, restenosis, ROS production, silica-based matrix, porous drug delivery system

## Abstract

In this study we present the porous silica-based material that can be used for in situ drug delivery, offering effective supply of active compounds regardless its water solubility. To demonstrate usability of this new material, three silica-based materials with different pore size distribution as a matrix for doping with Photolon (Ph) and Protoporphyrin IX (PPIX) photosensitizers, were prepared. These matrices can be used for coating cardiovascular stents used for treatment of the coronary artery disease and enable intravascular photodynamic therapy (PDT), which can modulate the vascular response to injury caused by stent implantation—procedure that should be thought as an alternative for drug eluting stent. The FTIR spectroscopic analysis confirmed that all studied matrices have been successfully functionalized with the target photosensitizers. Atomic force microscopy revealed that resulting photoactive matrices were very smooth, which can limit the implantation damage and reduce the risk of restenosis. No viability loss of human peripheral blood lymphocytes and no erythrocyte hemolysis upon prolonged incubations on matrices indicated good biocompatibility of designed materials. The suitability of photoactive surfaces for PDT was tested in two cell lines relevant to stent implantation: vascular endothelial cells (HUVECs) and vascular smooth muscle cells (VSMC). It was demonstrated that 2 h incubation on the silica matrices was sufficient for uptake of the encapsulated photosensitizers. Moreover, the amount of the absorbed photosensitizer was sufficient for induction of a phototoxic reaction as shown by a rise of the reactive oxygen species in photosensitized VSMC. On the other hand, limited reactive oxygen species (ROS) induction in HUVECs in our experimental set up suggests that the proposed method of PDT may be less harmful for the endothelial cells and may decrease a risk of the restenosis. Presented data clearly demonstrate that porous silica-based matrices are capable of in situ delivery of photosensitizer for PDT of VSMC.

## 1. Introduction

Percutaneous coronary interventions (PCI) involving stent implantation are routinely used for the treatment of coronary artery disease. Proliferation and migration of tunica media layer (middle coat) smooth muscle cells (SMC) in response to the vascular wall injury, are essential events leading to the subsequent neointimal thickening, which eventually causes vessel narrowing [1]. Therefore, three generations of drug eluting stents (DES) stents were designed with the ability to block SMC proliferation [2]. DES elute antiproliferative drugs into the surrounding tissue, thus indiscriminately block vascular cells [3]. Although, blocking SMC proliferation is important for hindering intimal hyperplasia, sustained growth of endothelial cells (EC) is essential for successful vascular repair. Moreover, the endothelial cells play an important role in the regulation of the vascular tone and inhibiting inflammation and thrombus formation. While modern DES successfully prevents neointimal proliferation, the presence of unspecific anti-proliferative drugs impairs the reendothelialization, which increases the risk of in-stent thrombosis [4]. Therefore, the new approach to design stents is needed with a rationale to block neointimal formation, while preserving endothelial function.

Photodynamic therapy (PDT) was shown to modulate the vascular response to injury caused by PCI [5,6]. PDT involves a photosensitizing agent (PS) that accumulates selectively in the target cells and its illumination with a low-power laser light of an appropriate wavelength to generate cytotoxic reactive oxygen species (ROS). The main role of PDT in the treatment of restenosis is to generate ROS that will interfere with SMC survival and consequent remodeling process [7]. PDT has been shown to induce redox-sensitive apoptosis of vascular cells [8,9]. In our previous in vitro study, we showed that chlorin e6 might be successfully used to induce apoptosis of smooth muscle cells [10] in PDT conditions. PDT also mediates the inhibition of transforming growth factor beta, inducer of endothelial cell apoptosis, and thus stimulates endothelial cell growth [11]. As such, PDT can contribute to facilitated endothelialization and oppose SMC proliferation and migration, thereby preventing neointimal hyperplasia. Experimental evidence from in vivo studies suggests that PDT can be also applied to prevent restenosis [12,13]. Intraarterial PDT at the time of balloon vessel injury was shown to inhibit intimal hyperplasia in animal model [14]. Despite that PDT is a one-time procedure it can induce biochemical processes leading to long-term inhibition of intimal hyperplasia [15].

Delivery of the soluble form of PS and its good selectivity are crucial for successful PDT. Usually, for intravascular PDT administration of PS is performed via intravenous injections. PS is taken up and retained in the lesion and administered prior to exposure to irradiation. However, there are some well-known disadvantages of the systemic infusion of PS due to poor tissue selectivity [16]. It has been reported that PS can reach and destroy blood cells, even when not administered intravenously [17,18]. That, in combination with residual dark toxicity of photosensitizers [19] can be a source of serious off target effects. To overcome these problems, novel delivery strategies have been developed. One of them uses liposomes [20], which can increase the specificity of delivery. Yet, the most effective and safe method is a local administration of high dose of PS. This approach would help to decrease the overall dose of the drug, as well as shorten the time of exposure and support the feasibility of PDT under clinical conditions. The feasibility of local delivery of liposomal verteporfin by balloon catheters was demonstrated in the ex vivo experimental studies. Effectiveness of such a strategy was further confirmed by detection of apoptosis of atherosclerotic plaque cells after photodynamic treatment with a diode laser 690 nm [21].

For our study we have chosen two widely used photosensitizers: Photolon (Ph) and Protoporphyrin IX (PPIX). Both drugs were already successively used in the cardiovascular PDT [22,23,24,25,26,27,28].

Our goal was to create a material suited for use as coating of cardiovascular stent. Such a material can serve as a protective layer as well as a source of PS. In contrast to DES, which is designed for very slow releases of drug, in this study we propose delivery of PS directly to cardiovascular plaque. Such an approach eliminates the need for systemic administration of a photo-sensitizer, which is the main source of side effects in PDT.

## 2. Materials and Methods

### 2.1. Photoactive Matrix Preparation

The sol–gel materials were prepared from silica precursor named TEOS (tetraethoxysilan 98%, Sigma-Aldrich, Saint Louis, MI, USA) mixed with 96% ethyl alcohol (Polish Chemicals) with addition of 36% HCl (Polish Chemicals) as a catalyst and Triton X-100 (Sigma-Aldrich). The mixture was stirred for 4 h with the speed of 400 rot/min at the room temperature. Diverse material samples were produced with the various ratio of number of TEOS moles to the number of ethyl alcohol moles. The suitable amount of solvent was used, thus to obtain the required materials, named as M1, M2, and M3. These materials were stored for one month in controlled temperature and humidity conditions (temperature 22 °C, humidity 68%). Then the morphology of solid samples was studied (pore size distribution and X-ray micrograph distribution).

For atomic force microscopy (AFM) studies as well as Attenuated Total Reflectance with Fourier Transform Infrared Spectroscopy (FTIR-ATR) and UV-vis spectroscopy, Photolon (18-carboxy-20-(carboxymethyl)-8-ethenyl-13-ethyl-2,3-dihydro-3,3,12,17-tetramethyl-21H, 23H-porphin-2-propionicacid) (Belmedpreparaty, Minsk, Belarus) and Protoporphyrin IX dimethylester (dimethyl-8,13-divinyl-3,7,12,17-tetramethyl-21H, 23H-porphine-2,18-dipropionate) (Fluka Honeywell, Charlotte, NC, USA) were used as photosensitizers (PS). Stock solutions 0.5% of photosensitive dyes were prepared by dissolving the photosensitizer in ethyl alcohol. The adequate amounts of stock solutions were added to 1 mL of the fresh prepared silica sol in order to get the final concentrations of PS depicted by Table 1.

### 2.2. Pore Size Distribution Experimental Setup

Examination of the porous structure of solid samples was carried out quantitatively by mercury porosimetry. Thermo Scientific Pascal 440 porosimeter (Thermo Fisher Scientific, Waltham, MA, USA) provides information about pore size distribution, particle size distribution, bulk density, and specific surface for most porous solids. For this study the structural parameters, such as relative pore volume and pore radius distribution, were determined. Each sample M1, M2, and M3 mass was 0.15 g [29].

### 2.3. Analysis of the Pore Size Distribution

The obtained results of the mercury porosimetry measurements in the form of dependence of the relative pore volume on the pore size, were used to characterize the pore size distribution in each analyzed material (M1, M2, and M3). A non-standard estimation method was developed and applied. The raw data produced by Pascal 440 device were fitted by Gaussian functions to evaluate the pore size distribution. For estimation of the goodness of fit the following measures were used: R^2^ statistic, the degrees of freedom adjusted-R^2^ statistic and RMSE (root mean squared error).

The R^2^ statistic measures with what accuracy the applied fit has in explaining the variation of the analyzed data. It is defined as the square of the correlation between the observed values and the predicted values from the fit. It can take values from 0 to 1, where values closer to 1 are indicating that a greater proportion of variance is accounted for by the applied fitting model.

In some cases, the increase of the number of fitted coefficients in the model can increase the value of the R^2^ statistic, but without the practical improvement of the fit. To omit this problem, the additional degrees of freedom adjusted-R^2^ statistic, were applied. This approach uses the R^2^ statistic and adjusts it basing on the residual degrees of freedom, which are defined as the difference between the number of the observation value and the fitted coefficients estimated from the observation values. As R^2^ statistic, adjusted-R^2^ statistic takes values from 0 to 1 and increases only when the additional fitted coefficients will improve the prediction model. Additionally, the RMSE measuring differences between observation values predicted by a fitting model and observed values, were applied. This parameter is estimating the standard deviation of the random component in the data. When RMSE takes values closer to 0, it indicates that the fit will be more useful for prediction of the observation’s values.

### 2.4. Light Power and Energy Densities Measurements

The energy density H and power density E were set up experimentally because the producer suggestions were tissues established only [30]. Power density was measured by integrating sphere photodiode power sensor (S142C, 350–1000 nm, 1uW-5W, Thorlabs, Newton, NJ, USA) and Compact Power and Energy Meter Console (PM100D, Thorlabs, Newton, NJ, USA).

### 2.5. Light Source

The used irradiation setup included laser diode (λ = 655 nm) with adjustable power control unit coupled into the optical fiber system (FC-655nm-1W-15070826, Changchun New Industries Optoelectronics Tech. Co., Ltd., Jilin, China) [31].

### 2.6. Energy-Dispersive X-Ray Spectroscopy (EDS)

The structure of coatings was examined by scanning electron microscopy Hitachi S-4000 (Hitachi, Omuka, Japan) equipped with energy dispersive X-ray system. Prior to the examination, the samples were stored in a desiccator. The spectra were recorded by the following conditions: accelerating voltage 20.0 kV, beam current 100,000 nA, magnification 1000, live time 77 s, preset time 0, Nb channels 2048, Ev/Channel 10, offset 0 keV, width 20 keV, whereas the geometry was as follows: tilt angle 0, azimuth angle 0, take-off angle 33 deg.

### 2.7. Atomic Force Microscopy

AFM images were acquired in the tapping mode using a Nanoscope IIId scanning probe microscope with Extender Module (Bruker, Billerica, MA, USA) in the dynamic modus. An active vibration isolation platform was applied. Olympus etched silicon cantilevers were used with a typical resonance frequency in the range of 200–400 kHz and a spring constant of 42 N/m. The set-point amplitude of the cantilever was maintained by the feedback circuitry to 80% of the free oscillation amplitude of the cantilever. All samples were measured at room temperature in air. The sample was first adjusted with an optical light microscope (Nanoscope, Optical Viewing System).

For this study, mica plates (Nanowords) were coated by undoped and Photolon doped sol–gel matrices, as well as undoped and Protoporphyrin IX (PPIX) doped sol–gel layers. The distribution of PS molecules on the coating surface was analyzed.

### 2.8. Spectrophotometric Measurements

All absorption spectra were recorded in wavelength range 200–1100 nm by the means of the AVA-Spec 3648 spectrophotometer (Avantes, Apeldoorn, The Netherlands) equipped with deuterium-halogen lamp (Avalight-DH-S-BAL, Avantes) as the light source. As the excitation source in luminescence measurement the continuous wave semiconductor laser λ = 405 nm (TOPGaN, Warsaw, Poland) was used. For spectroscopic studies the PSs doped M1, M2, and M3 materials were measured in standard UV cuvette. After that materials were deposited on a round coverslip glass with a diameter of 12 mm (20 µL per slide). An important step in this process was the deposition of a thin and uniform layer of silica material in which the PSs are immobilized.

### 2.9. Fourier Transform Infrared Spectroscopy (FTIR)

316 L stainless steel functionalized surfaces (10 mm in diameter) were characterized with FTIR-ATR spectroscopy. Spectra were recorded using attenuated total reflection method with Nicolet 6700 FTIR spectrometer (Thermo Fisher Scientific,) and Diamond Top-Plate of the Golden Gate MK II ATR Accessory (Pike, Fitchburg, WI, USA). Samples were placed directly on the diamond at steady temperature of 21 ± 1 °C. Spectra were obtained as a mean of 32 scans with the resolution of 4 cm^−1^ using the OMNIC software (version 7.4, Thermo Fisher Scientific) at the standard analysis field. Further, spectral analysis and processing was carried out with Origin software.

### 2.10. Cell Culture Conditions and Materials

Human peripheral blood mononuclear cells (PBMCs) were cultured in RPMI medium supplemented with 10% fetal bovine serum (FBS; Gibco, Thermo Fisher Scientific). Blood samples were collected from healthy consent, volunteers, with agreement with local regulations and Ethical Board permit no. KB-158/2010. Human umbilical vein endothelial cells (HUVEC) and human aorta/smooth muscle cells (T/G HA-VSMC) were maintained in F12 K medium supplemented with 2 mM l-glutamate (Gibco) and 0.1 mg/mL heparin (Sigma-Aldrich), 0.03 mg/mL endothelial cell growth supplement (Sigma-Aldrich), and 10% FCS. HUVEC, T/G HA-VSMC cells and PBMCs were cultured on flasks and 24, 48, or 96-well plates (Corning Costar) at 37 °C in a humidified atmosphere of 5% CO_2_.

### 2.11. Confocal Laser Microscopy

For PS uptake HUVEC cells were seeded on a 96 well plate (12 × 10^3^ cells per well) coated with photosensitizer-loaded silica sol–gel carriers. After 4 h incubation the probes were illuminated with light wavelength 655 nm (power density E = 420 mW/cm^2^, energy density H = 12.5 J/cm^2^) and incubated 1 h before fixation. Uptake of PS by PBMCs was examined after 45 min incubation with test compound surfaces. Briefly after isolation, lymphocytes were seeded on 24 well PS-loaded silica sol–gel carriers plate. Next some of the probes were illuminated with light wavelength 655 nm (E = 420 mW/cm^2^, H = 12.5 J/cm^2^). After each treatment cells were washed twice with PBS (phosphate-buffered saline solution, Ludwik Hirszfeld Institute of Immunology and Experimental Therapy) and fixed with 4% paraformaldehyde (Sigma-Aldrich, Saint Louis, MO, USA) in PBS, at 37 °C for 30 min. Then cells were washed three times with PBS. The confocal images of cells were collected on confocal laser microscope Leica TCS SPE using a 63× high numerical-aperture oil immersion objective. Image size was set to 512 pixels × 512 pixels. For excitation of PS laser line at 405 nm was operated at 20% of maximum power. Optimal gain settings 1260 V was used.

### 2.12. Biocompatibility and Toxicity Studies

For hemolysis assay 8 mL of peripheral blood was obtain from anonymous human donor and drawn into K2-EDTA-coated Vacutainer tubes (Becton Dickinson, Franklin Lakes, NJ, USA). Erythrocytes were prepared as described by Evans et al. [32], with minor modifications. Shortly, blood cells were centrifuged (450 × *g*; 5 min; 14 °C) in 15 mL conical tubes (Sarstedt, Nümbrecht, Germany) and washed in 0.9% NaCl or PBS pH 7.4. Cells were extracted in 15 mL conical tubes (Sarstedt) and diluted in 50 mL conical tube (Falcon, Corning Costar Inc., Corning, New York, NJ, USA).

Hemolysis assay was performed at black 96 well plates previously coated with the test compounds. Red blood cells were diluted 1:49 (*v*/*v*) in PBS pH 7.4 and 190 μL was added to each well. After 1 h incubation at 37 °C, the cells were centrifuged at 450 × *g* for 10 min at 14 °C and 100 μL of supernatant was carefully transferred to a transparent 96 well plate (Corning Costar Inc.). The absorbance at 450 nm corresponding to hemoglobin release from lysed erythrocytes was measured by Thermo LabSystems Multiskan RC Microplate Reader (Thermo Fisher Scientific). Erythrocytes incubated with 1% Triton X-100 (Sigma-Aldrich) served as positive control (100% hemolysis). Result below 5% was considered as negative (no hemolysis).

Peripheral blood lymphocytes were isolated from blood using Lymphocyte Separation Medium 1077 (Cytogen, Wetzlar, Germany) according to the supplier’s instruction. Lymphocytes at density 1 × 10^6^ were seeded on 24-well plate (Corning Costar Inc.) on coated surfaces and incubated (37 °C, 5% CO_2_). For assessment photo-toxicity cells were illuminated (λ = 655 nm, E = 420 mW/cm^2^, H = 12.5 J/cm^2^) and removed to fresh plates and cultured for 48 h. Next, the apoptosis was evaluated using the method of Nicoletti et al. [33], basing on detection of cells with sub-diploidal DNA content. Shortly, cells were harvested, washed (PBS), fixed with 70% ethanol and stained with propidium iodide (10 µg/mL in PBS). Washed cells were analyzed using FACSCalibur flow cytometer (Becton Dickinson). Cell debris was excluded from the analysis by electronic gating. The DNA content was evaluated on the basis of FL-3 histograms using Flowing Software 2.51 (Turku Bioscience, Turku, Finland). Apoptosis was quantified as the percentages of cells with hypodiploidal DNA content.

### 2.13. Intracellular Reactive Oxygen Species Level

Reactive oxygen species (ROS) concentration was assessed by measuring the 485-/535-nm fluorescence of H_2_DCFDA (6-carboxy-2′, 7′-dichlorodihydrofluorescein diacetate, di(acetoxymethyl ester); Molecular Probes, Thermo Fisher Scientific, Inc., Waltham, MA, USA. Briefly, cells were stained with 10 μM of H_2_DCFDA for 30 min in PBS with 10% FBS and washed twice with warm PBS with 2.5% FBS. Then HUVEC and T/G HA-VSMC cells were seeded on photoactive surfaces on 48-well plate at density 60 × 10^3^ cells per well, after 2 h the basal fluorescence was measured. Next, the cells were illuminated (λ = 655 nm, E = 420 mW/cm^2^, H = 12.5 J/cm^2^) and then fluorescence was measured again. Basal fluorescence and fluorescence collected from matrices without photosensitizer values were used as the background and was subtracted from all the samples after illumination.

## 3. Results

### 3.1. Analysis of the Pore Size Distribution and Energy Dispersive X-Ray Spectroscopy

The matrices with controlled porosity make silica materials highly attractive as the structural basis for a wide variety of technological applications, e.g., medical devices, therefore we report the effects of synthesis parameters on pores size distribution and the elemental composition.

On Figure 1a data from porosimeter together with fitted pores distributions are presented. Some of prepared silica sol–gel materials have heterogeneous pores distribution, and some contain a large majority of small pores. As one can see the preparation conditions used for M1 samples production resulted in small pores generation in the final solid matrix, whilst this effect was not observed in the other samples. The M1 material had 49% of pores diameter <5 nm. In the sample M2, the 29% pores diameters were in range 15–50 nm and 8% smaller pores in range 1–5 nm were noticeable. In the sample M3 there were 26% small pores (1–10 nm), 30% pores in range 10–50 nm, and also large pores were observed with the diameter ranging from 2000–5000 nm (17%).

The energy dispersive X-ray spectroscopy (EDS) was used to verify the chemical composition of the materials. The EDS results on Figure 1b revealed the presence of carbon, oxygen, and silicon elements with expected ratio, compatible with the reaction substrate. The highest amounts of oxygen and silicon were detected in the M1 material. In M2 and M3 materials the amount of oxygen and silicon decreased with increasing carbon amount. Carbon was the predominant element in materials M2 and M3 and its amount was increasing with the increase of ethyl alcohol volume derived from synthesis.

The raw data from mercury porosimetry was fitted by two Gaussian functions on a decaying exponential background using the nonlinear model, to estimate the pore size distribution of investigated materials. Obtained values of the measures used to evaluate the goodness of the fit are presented in Table 2. Based on the adjusted-R^2^ statistic it was shown that the used fitting models enabled the explanation of the 99.66%, 92.41%, and 93.66% of the total variation in the data obtained for M1, M2, and M3, respectively.

The fitted pore size distributions in each analyzed material are shown on Figure 1a. Some of the prepared silica sol–gel materials had non-homogenous pores distribution, and some contained a large majority of small pores. It appeared that a particular set of preparation conditions used for M1 samples favored the generation of small pores. Resulting M1 material consisted mainly of pores with a diameter in the range of 1–5 nm (49%) and 15–50 nm (10%). The material M2 exhibited a more heterogeneous structure with pores size distribution with two maxima for 15–50 nm (28%) and 3000–8000 nm (16%). The material M2 exhibited a more heterogeneous structure in pores size distribution with two maxima for 15–50 nm (28%) and 3000–8000 nm (50%). On the other hand, results of the analysis obtained for material M3 were indicating that it exhibited the most heterogeneous pore size distribution with the maxima for pore sizes in the range of 5–70 nm (10–14%) and 1000–8000 nm (5–10%) Interestingly, pores with the size ranging from 700 to 1000 nm were rare in all synthesized material. The silica material M1 had the lowest pore volume of all examined samples. In principle, the pore volume decreased with increasing solvent volume.

### 3.2. Atomic Force Microscopy Examination

The surface morphology of silica sol–gel materials was characterized by atomic force microscopy. The AFM images of M1, M2, and M3 sol–gel layers revealed that the different porosity of materials had limited impact on the surface composition and surfaces of all coatings were smooth and homogeneous (Figure 2) with roughness ranging from 1.2 to 1.5 nm. Addition of photosensitizers (Ph or PPIX) ensued in even more reduced roughness of all the surfaces. Resulting photoactive matrices were very smooth as the measured roughness profiles ranged from 0.5 to 1.0 nm.

### 3.3. Spectrophotometric Measurements

PS’s absorption and emission bands should not overlap with absorption or fluorescence bands of endogenous chromophores occurring in human body. Therefore, optical properties of PSs and PS-doped sol–gel carriers were determined by spectrophotometry. The absorption spectrum of Photolon (Ph) shows a strong band at 410 nm (Soret band) and weaker Q-bands at 528 nm, 595 nm, and 645 nm (Figure 3a). For Protoporphyrin IX (PPIX) absorption bands were observed at 403 nm, 420 nm (Soret band), and at 561 nm and 606 nm (Q-bands; Figure 3b).

The changes related to the type of material were firmly visualized by luminescence intensity in colloids (Figure 3c,d). Upon excitation at 405 nm, the emission spectrum of Ph showed a strong maximum at 670 nm observed in all colloids M1-Ph, M2-Ph, and M3-Ph. The highest luminescence signal was noticed for M3-Ph colloid, which contained the highest amount of ethyl alcohol and was related to the high Photolon solubility. 

In layers deposited onto the glass coverslip one could observe the strong PS-material interaction. At the same irradiation parameters (λ_ex_ = 405 nm) the strongest luminescence for M2-Ph, and lower for M3-Ph and M1-Ph, was observed (Figure 3e) and there was no band shift compared to Figure 3c (colloids).

For PPIX doped silica colloids luminescence maxima were found at 633 nm and 700 nm, and the signal was the highest for M1-PP IX sample. This might be related to the beginning of the structural changes in silica colloid leading to pores formation in the solid material. The predominant amount of small size pores in the material (<5 nm) favored the monomeric form of PSs. Aggregated PS are less photochemically active because of nonradiative deactivation of excited states, which shortens the fluorescence lifetime [34].

In the case of silica coatings doped with PP IX (Figure 3f) all bands were very wide. Except prominent and strong 633 nm and 672 nm bands, a completely new band appeared at 650 nm, and all other bands were widening. Widening of the 633 nm band was clearly noticed and this band dominated in M1-PP IX luminescence. For M2-PP IX and M3-PP IX band 672 nm was higher than 633 nm. At layers spectra M1-PP IX, M2-PP IX, and M3-PP IX one could recognize: right shoulder 700 nm (like in colloids), strong band 672 nm, between 672 and 633–648 nm band, strong band 633 nm (like in colloids), and left shoulder 620 nm. The reason of Q-bands excitation in PPIX luminescence spectra was probably induced by the association effect of PPIX due to the aggregation of porphyrins located in larger pores of the matrices. The same effect was observed by appearing “red shift” phenomena in the case of FTIR spectra of PPIX in matrices.

### 3.4. Spectroscopic Characterization of Functionalized Surfaces

FTIR spectroscopy was used to determine the structural characteristics of functionalized 316 L samples. Initially, spectra of Ph (Figure 4, top curve) and PPIX (Figure 4, bottom curve) were recorded to serve as the reference for a further sample study. For interpretation clarity spectrum has been divided into three specific areas. First area (I) ranging up to 1000 cm^−1^ is the most differentiating and full of various signals, most of which are very likely to be coming from bending vibrations of groups, such as (C–H), (C=C), and (N–H). Spectral region between 1000 cm^−1^ and 1300 cm^−1^, marked as II is filled with peaks attributed to stretching vibrations of amines (C–N), alcohols (C–O), carboxylic species (C–OH), some skeletal vibrations from (C–C), and, finally, of ester groups (C–O–C). The last described area is the least differentiating, however the dissimilarities are also easy to spot. In both spectra, slightly shifted signals of 1375 cm^−1^ and 1422 cm^−1^ are assigned to the bending vibrations of (C–H) groups and to skeletal stretches of (C=C). These signals are followed by an amine doublet, with maxima located at 1566 cm^−1^ and 1652 cm^−1^ (bottom curve, 1563 cm^−1^ and 1620 cm^−1^), that seems to differ in each spectrum regarding the signal intensity, most likely due to the exposure of (N–H) groups. Peaks located at 1669 cm^−1^ and 1712 cm^−1^ are present only in Protoporphyrin IX spectrum, they are most likely coming from stretching vibrations of imine groups (C=N) or skeletal (C=C) and (C=O) from ketonic species, respectively. Next in line are the signals located between 2700 and 3100 cm^−1^ assigned to stretches of (C–H) groups coming from various kinds of alkanic and alkenic species. This area is definitely more diverse in the case of Protoporphyrin IX spectrum, because there are at least six different local maxima, as opposed to a simple triplet in case of Ph. The last and perhaps the most prominent difference between the two spectra are the peaks located at 3271 cm^−1^ for Ph and at 3309 cm^−1^ for PPIX. Broad nature of this signal in Ph spectrum suggests that it is correlated with the stretching vibrations of hydroxyl (O–H), which is not the case for PPIX, because this compound simply lacks those groups. However, the sharp character of the PPIX 3309 cm^−1^ signal leads to a conclusion that it corresponds to amine stretches of (N–H) groups.

The spectra of surfaces modified with M1, M2, and M3 were evaluated (Figure 5a–c respectively). The characteristic signals of all spectra, before the addition of porphyrins were very much alike, due to the coatings being different only in the molar ratios of ethanol and TEOS. Before the addition of porphyrins, spectra varied mostly in the intensity of each peak and in the area of the most prominent peak, located at 1100 cm^−1^ of II region. In the case of M1 and M2 this band was quite uniform and represented the signal from stretching vibrations of (Si–O–Si), whereas in M3 it noticeably differentiated into two maxima at 1110 cm^−1^ for (Si–O–Si) and at 1157 cm^−1^ for carboxylic stretches of (C–OH). Otherwise, the signals looked very similar in all three regions. However, slight spectral changes became visible after the addition of Ph, when a small doublet signal emerged at 1620 cm^−1^ suggesting the presence of bending vibrations of amine (N–H), most likely coming from the addition of photosensitizers. The most visible difference seemed to be the flattening of a broad 3300 cm^−1^ signal, coming from stretching vibrations of hydroxyl (O–H) groups. At the same time a steep slope-like signal was formed at 3350 cm^−1^, which corresponded to amine stretches of (N–H) bonds, again confirming the presence of Ph. The other difference, noticeable after the addition of this porphyrin was the slight shift and change of shape of the signal of the highest intensity, located in II region at 1100–1200 cm^−1^. In the case of M1 it emerged as a doublet similar to M3 before the addition of porphyrins, whereas in M2 and M3 it shifted a little to the lower wavelength values. The addition of Ph also seemed to smoothen the signals located in the area ranging from 1250 cm^−1^ to 1510 cm^−1^, which was mostly responsible for bending vibrations of (C–H) groups. When it came to photoactive surfaces, the most visible spectral differences appeared in the slight shifts of the 1100 cm^−1^ signal and the appearance of the amine doublet at 1620 cm^−1^, also coming from stretching vibrations of (N–H) groups. In the case of Protoporphyrin IX addition, the regions I and III stayed relatively unchanged.

Interestingly, all matrices doped with PPIX seemed to exhibit the “red shift” effect of bending vibrations of (N–H) groups, meaning that the signals responsible for those groups tended to shift slightly towards the lower wavenumbers (area marked with yellow circle in each PPIX modified spectra of Figure 5). The shifts were subtle and they progressed along with the size of the pores of the matrices. For M1, which had the smallest pores, the amine doublet was characterized by two peaks, located at 1615 cm^−1^ and 1649 cm^−1^. In the case of matrix M2, those two peaks were slightly shifted and their values were 1611 cm^−1^ and 1638 cm^−1^. However, the biggest shift, with regard to M1, was observable for M3, where the amine signal was located at 1608 cm^−1^ and 1632 cm^−1^. This phenomenon, known as the “red shift”, was most likely the indicator of association effect, meaning that (N–H) groups of PPIX were closing in on each other due to the deformation or aggregation of porphyrins located in pores of the matrices. This effect was also observable, and thus confirmed, in luminescence spectra.

### 3.5. Blood Biocompatibility and Toxicity Studies

Two models corresponding to predicted implementation of developed photoactive materials in photodynamic therapy of vascular lesions were used. As contacts of red blood cells (RBC) with some silica-based materials may result in disruption of RBC membrane due to hydrophobic [35] or electrostatic interactions with material [36,37,38] we checked, whether incubation of human RBC on this silica-based surface will affect the RBC integrity. RBC hemolysis is reflected by hemoglobin release and can be detected as an increase of the supernatant absorbance at 450 nm. In all studied samples, after 1 h incubation on surfaces coated with silica-based photoactive surfaces, we did not observe any hemolysis-induced hemoglobin release (more than 5% Triton X-100 of positive control). Additionally, we studied the impact of silica based photoactive surfaces on the viability of human peripheral blood mononuclear cells (PBMCs). For this purpose, the assay, which detects apoptotic cells on the basis of DNA content, was used. Even as prolonged as 48 h culture of peripheral blood lymphocytes on tested surfaces resulted in apoptosis less than 10% of the PBMCs, showing that all tested surfaces exhibit only very little dark toxicity.

With the lack of dark cytotoxic effect, we used the same assay for testing phototoxic properties of silica-based photoactive matrices in PBMCs. As in the previous studies cells were preincubated on photosensitizing surfaces for as long as 24 h (Figure 6). All the materials exhibit similar and negligible dark toxicity, but they varied in induction of phototoxicity of PBMCs. The illumination of materials did not result in an increase of apoptosis of PBMCs incubated on photosensitizer-free surfaces or surfaces modified with Ph. Materials doped with PPIX were able to photosensitize some part of PBMCs. It should be noted that the percentage of apoptosis about 20% could be regarded as relatively low in the context prolonged exposure on photosensitizing surface, which were not expected in the planed application. It shows that in implementation related conditions the materials should express little off-target activity.

### 3.6. PS Uptake and ROS Production

For assessment of biological activity two cell lines relevant to the proposed application were selected: human umbilical vein endothelial cells (HUVECs) and vascular smooth muscle cells (VSMC). Cellular uptake of photosensitizers encapsulated in silica matrix carriers was studied using confocal microscopy after incubation of HUVECs and VSMC on Ph- or PPIX-loaded M1, M2, and M3 photoactive materials. Fluorescence of Ph or PPIX was observed in all probes, showing that both photosensitizers could be efficiently taken up by the cells from M1, M2, and M3 surfaces in 2 h (Figure 7, for control image with undoped surfaces see Appendix A). Inside the cells, photosensitizers were mainly located in the cytoplasm, but some nuclear staining also could be observed.

The direct outcome of photodecomposition is the generation of ROS inside the cells. This event usually starts a cascade of reactions leading to programmed cell death. In this regard, we checked the sufficiency of cellular uptake of PSs to induce ROS production upon illumination. Both the HUVECs and the VSMC were incubated on the M1, M2, and M3 materials loaded with Ph or PPIX and illuminated with 655 nm light beam corresponding to PDT window of Ph and PPIX. The fluorescence collected from cells incubated on matrices without a photosensitizer, was negligible and ranged from 154 to 225 RLU. In the case of matrices with photosensitizers, the M1 material was most effective for induction of ROS in the both cell lines, regardless the photosensitizer. Two other materials were less efficient, especially in case of PPIX. Importantly, ROS production was markedly higher in vascular smooth muscle cells than in endothelial cells (Figure 8). To further confirm that HUVEC stayed intact we performed confocal laser microscopic examination (Figure 7).

## 4. Conclusions

As cardiovascular stents are commonly used during PCI procedures, PS release could be accomplished from the surface of the stent coating. At the same time immobilization of PS may reduce its systemic toxicity by limited release from the matrix. This approach combines encapsulation of PS into the interconnected porous matrix with procedure of the catheter accessible lesion illumination. We show that cardiovascular stents can be upgraded into PDT devices by deposition of PS loaded layer onto the inner surface. Our paper presented in vitro studies concerning PSs contained coatings that might be deposited on the medical device. During the PDT, the reactive oxygen species could be generated not from the surface of the layer but from interconnected pores network of deposited coating still containing entrapped PS molecules.

The preparation of silica-based materials via sol–gel technology was affected by variations in the synthesis conditions. The materials were properly obtained if a catalyst (hydrochloride acid) was used, otherwise a non-durable or opaque material, indicating the formation of the internal stresses, was obtained. It was found that the gelling time increased as the dilution increased (higher alkoxide/alcohol ratio), and the gel formed was transparent and rigid [39]. During the ageing step, the excess of alcohol was expulsed from the gel, with a consequent shrinking. In general, the shrinking was more pronounced in gels synthesized using ethyl alcohol than using water as a solvent, suggesting the formation of denser and less porous gels [40].

We observed the best optical properties (the highest transmittance, the lowest absorbance, and no scattering in visible region) for M1 material. Using a mercury porosimetry it was stated that this particular material had monomodal pores distribution. Materials M1 prepared with TEOS to ethyl alcohol molar ratio 20 (Table 1) was used before for medical sensors construction [41], for production of the applicators for laser induced interstitial therapy [42], and for intravascular photodynamic therapy as stent coating [43]. Furthermore, the FTIR spectroscopic analysis confirmed that all studied surfaces were successfully functionalized with the target photosensitizers.

It is worth to mention that we were able to prepare photoactive materials regardless of differences in chemical properties of photosensitizers. Both, well water soluble Ph and low water soluble PPIX were successfully encapsulated in silica based porous matrices.

Low material roughness results in reduced friction between the rough implant surface and the blood vessel wall during stent placement [44]. This limits the damage of the inner wall of the artery and reduces the risk of restenosis. Maximum absorption of PPIX and Ph does not overlap maximum absorption of natural chromophores occurring in human body. Encapsulation of PS into the matrix provides drug delivery straight to the targeted cells. Photoactive matrices are biocompatible. The hemocompatibility studies were performed on the human blood cells. We show that the photosensitizer free as well as photosensitizer doped surfaces had no activity to induce the hemolysis of erythrocytes. The surfaces did not exhibit any dark toxicity and very low phototoxicity even after 24-long photosensitization. Thus, our newly developed material that could be regarded as promising photosensitizing modifier of medical devices in the treatment vascular lesions or atherosclerotic plaque.

The biological studies show that even short incubation of the SMCs on the photoactive matrices could result in PS uptake sufficient for induction of phototoxic reaction upon illumination measured as ROS induction. The uptake was detectable in both smooth muscle cells and endothelial cells. The photosensitizing surfaces varied in the activity, but the M1 material was the most efficient in induction of ROS. As this material had the smallest pores and lowest pore volume ratio it strongly suggests that the presence of small pores and/or low pore volume ratio was important for overall biological activity of obtained materials. It clearly indicates that M1 material was best for in situ delivery of a photosensitizer for PDT using Ph or PPIX. It can be expected that different pore size or pore volume ratio would be required by different drugs. It seems to be contradictory in view of a poor release of various drugs from porous matrixes with small pores [45,46], but on the other hand, more sustained release of drug from material with smaller pores may be beneficial for successful photosensitization.

Although, it is known that different cell subsets can differently response to PDT [45], in this case the lower sensitivity of endothelial cell line to PDT can be a result of higher expression of heme oxygenase-1 that protect cells from oxidative cells [46,47] and along with other factors can affect the stability of photosensitizers. This result indicates some selectivity of PDT towards the SMC line. We also observed, that Photolon-doped layers were more effective. Similarly, the differences could result from the fact, that the cells could better handle naturally occurring Protoporphyrin IX, than semi-synthetic PVP-modified chlorin e6, showing that Photolon was a more active photosensitizer in studied experimental conditions.

In clinical conditions, at the site of angioplasty, EC cells were absent (balloon/stent denudation), while SMC became activated and proliferated at a high rate. Thus, targeting the SMC proliferation is important for hindering intimal hyperplasia. On the other hand, proper reendothelialization is important for successful vascular repair decreasing the risk of in-stent thrombosis. Here, we show that PDT was able to discriminate between the different cellular subsets and could be more effective in smooth muscle cells than in the endothelial cells. It made the in situ PDT even more valuable therapeutic option for treatment of cardiovascular disease.

We designed porous and smooth silica-based matrices that could be successfully used for delivery of Photolon and PPIX photosensitizers. Differences in physiochemical properties of these two photosensitizers revealed the extent of the universality designed material. All tested materials were biocompatible and capable of in situ delivery of PS for PDT of VSMC.

## Figures and Tables

**Figure 1 materials-12-04110-f001:**
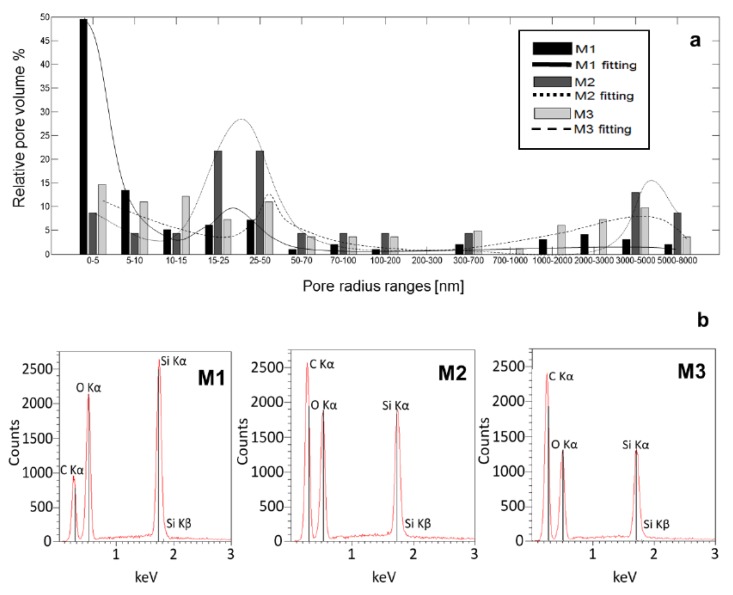
Pores distribution with fitted p.d.f. (**a**) Each vertical bar represents percentage of overall volume of pores for pores with given range of radius and X-ray microprobe distribution (**b**) for M1, M2, and M3 respectively.

**Figure 2 materials-12-04110-f002:**
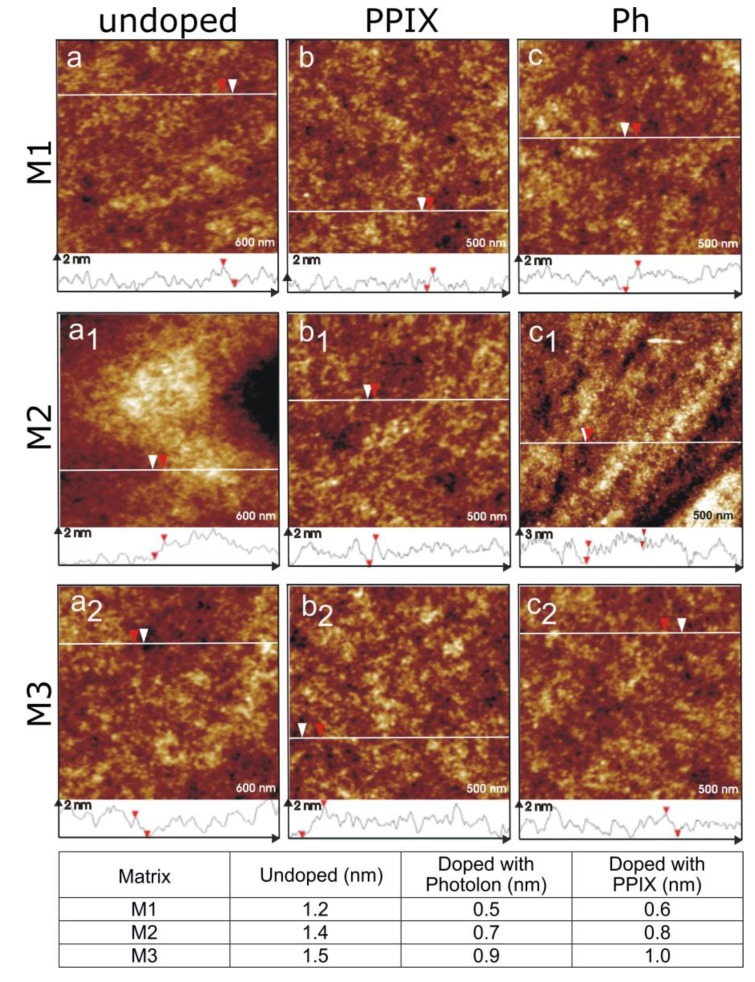
Atomic force microscopy images of (**a**,**a1**,**a2**) undoped silica sol–gel layers M1, M2, and M3 on mica respectively, (**b**,**b1**,**b2**) sol–gel layers M1, M2, and M3 doped with Protoporphyrin IX (PPIX), and (**c,c1,c2**) sol–gel layers M1, M2, and M3 doped with Photolon; roughness of the sol–gel layers calculated using the AFM technique.

**Figure 3 materials-12-04110-f003:**
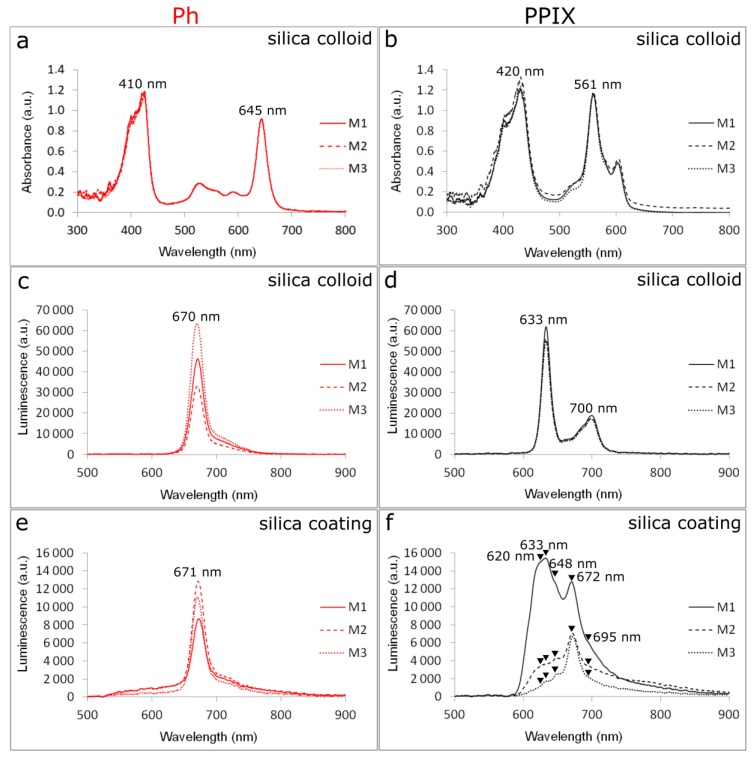
Absorption bands of (**a**) Photolon and (**b**) PPIX in silica colloids, luminescence intensity of (**c**) Photolon and (**d**) PPIX in silica colloids, and luminescence intensity of (**e**) Photolon and (**f**) PPIX in silica coatings; final concentration of PS is 0.01%.

**Figure 4 materials-12-04110-f004:**
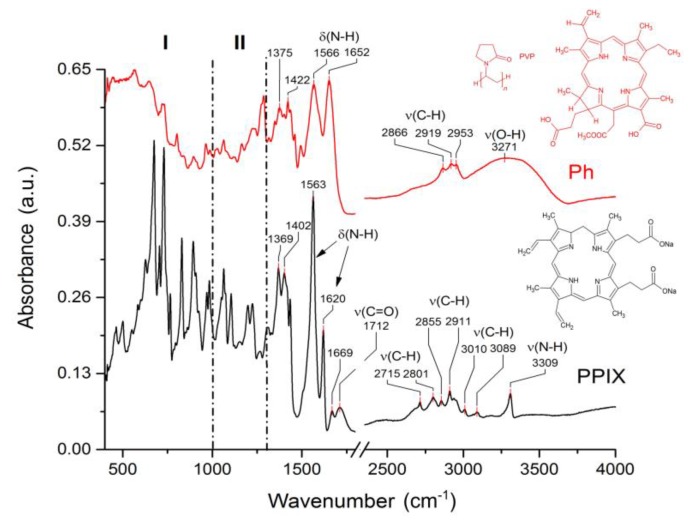
Attenuated total reflection Fourier transform infrared spectra of porphyrins with characteristic signals marked; Top panel (red line) represents the structure and spectrum of Photolon, bottom panel (black line) shows the structure and spectrum of Protoporphyrin IX.

**Figure 5 materials-12-04110-f005:**
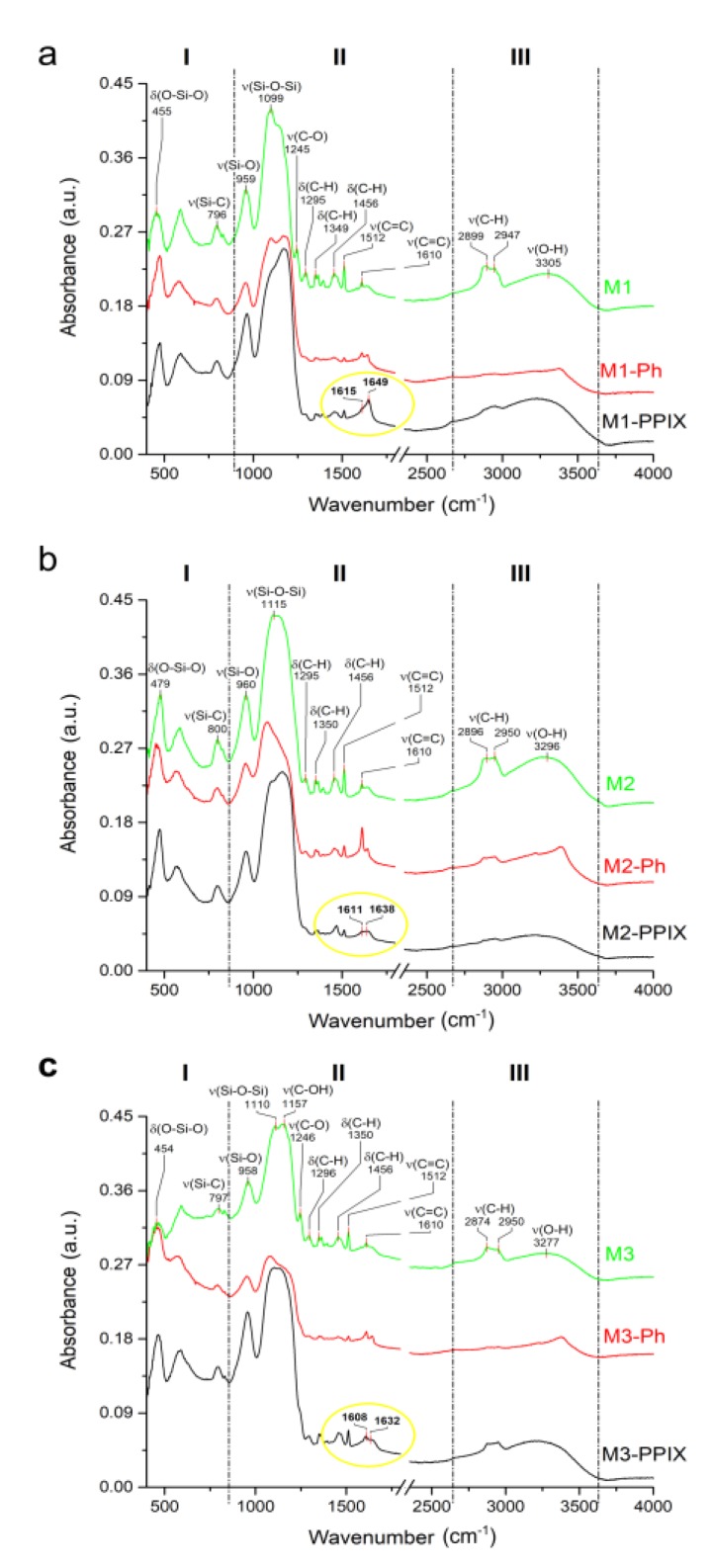
FTIR-ATR spectra of 316 L surfaces functionalized with: (**a**) M1 doped with porphyrins, (**b**) M2 doped with porphyrins, and (**c**) M3 doped with porphyrins; in each image, top curve (green) is the spectrum of porous material before the addition of any porphyrin, central curve (red) represents spectrum of porous sample with the addition of Photolon, while the bottom line (black) demonstrates the spectrum of surface with Protoporphyrin IX; and yellow circle represents the area of δ(N-H) groups and its progressing “red shift” observed for all PPIX doped matrices.

**Figure 6 materials-12-04110-f006:**
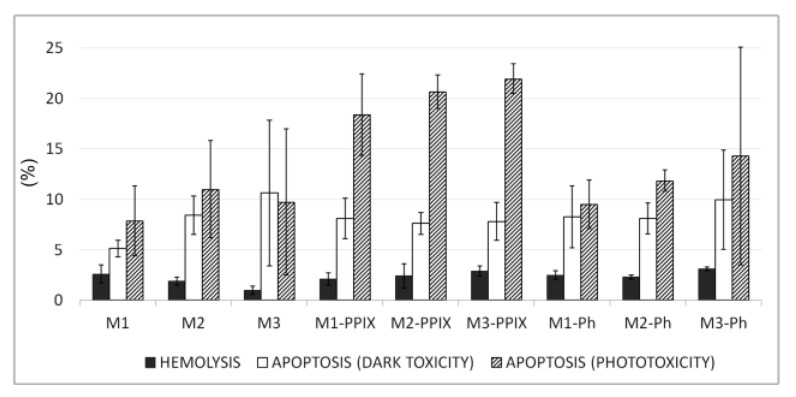
Biocompatibility and phototoxicity of the layers M1, M2, and M3 doped with PPIX and Ph. The average ± standard deviation is shown on the graph.

**Figure 7 materials-12-04110-f007:**
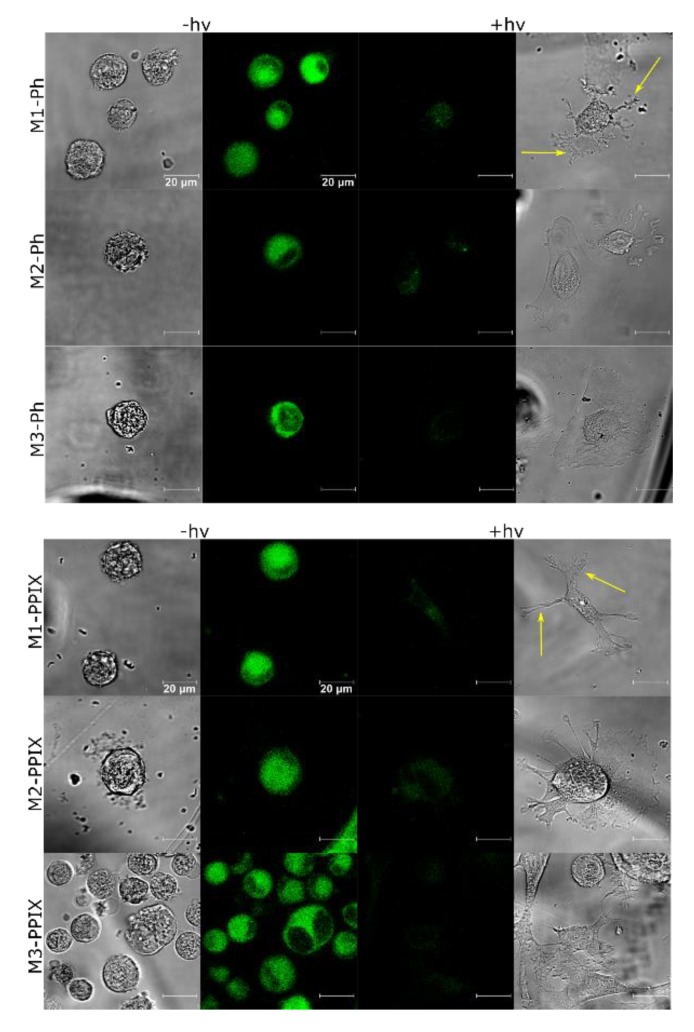
Confocal and bright field images of HUVEC incubated on sol–gel layers M1, M2, and M3 doped with Photolon and PPIX before (left panel) and after (right panel) PDT. Cells with pseudopodia (arrows).

**Figure 8 materials-12-04110-f008:**
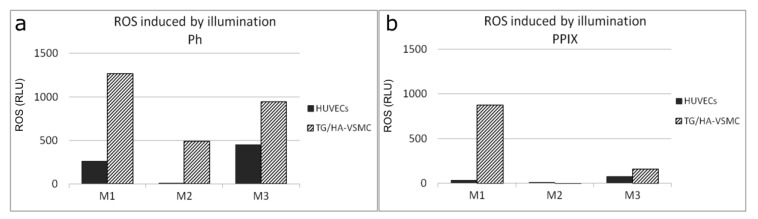
Relative fluorescence of reactive oxygen species (ROS) induced by illumination with 655 diode laser in TG/HA-VSMC and HUVECs incubated with M1, M2, M3 matrices doped with (**a**) Photolon and (**b**) PPIX.

**Table 1 materials-12-04110-t001:** The number of the substrate moles in M1, M2, and M3 coatings and the photosensitizing agents (PSs) doped M1-Ph, M2-Ph, M3-Ph, M1-PP IX, M2-PP IX, and M3-PP IX.

	# C_2_H_5_OH (mol)	# TEOS (mol)	# C_2_H_5_OH/# TEOS	# Ph (mol)	# PP IX (mol)
M1	0.288 × 10^−3^	14.38 × 10^−6^	20	16.7 × 10^−12^	17.4 × 10^−12^
M2	0.309 × 10^−3^	9.46 × 10^−6^	32		
M3	0.313 × 10^−3^	7.82 × 10^−6^	40		

**Table 2 materials-12-04110-t002:** The comparison of the goodness of the fit of the pore size distribution for M1, M2, and M3 materials based on used measures.

	M1	M2	M3
R^2^ statistic	0.9966	0.9241	0.9366
Adjusted-R^2^	0.9947	0.9034	0.9121
RMSE	4.181	6.509	3.616

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
