# Peer review of "Photoactive Pore Matrix for In Situ Delivery of a Photosensitizer in Vascular Smooth Muscle Cells Selective PDT"

_materials, 2019, doi:10.3390/ma12244110_

Round 1
Reviewer 1 Report
The manuscript submitted by Wawrzynska et al. reported an interesting approach for the fabrication of sol-gel silica matrices for an in-situ delivery of photolon and protoporphyrin photosensitizers. The obtained three types of materials (M1-M3) were characterized by UV-Vis spectroscopy, FTIR, AFM, ect. The application of these developed systems was shown by releasing the photosensitizers to HUVECs cells by photodynamic therapy.
Did the author quantify how much of photolon and protoporphyrin photosensitizers incorporated into the silica matrices? Also how much of them released and retained in the matrices should be measured. Given the largest pore size for all materials, why AFM does not show any such features? The author should comment on it. Line 368-370: the red shift of the peak are in correlation with the pore size of the materials. Even if this true, the shift should be higher for M2 material which showed a pore size distribution of 3000-8000 nm, compared with M3. This should be explained with proper citation. Line 151: can the author use EDX to determine the structure of coating or chemical composition of the coating? The aim of the paper (line: 84-92) is vaguely described and it should be clearly pointed out in the revised paper. In addition, the conclusions are very lengthy, and it could be concisely described. Line 44: what is DES?? Line 104-108: I could not understand why photolon and protoporphyrin was used in particular for IR, AFM and UV-Vis? This should be explained. The experimental part, including the preparation of sample preparation for each technique used in this paper. This is not clearly described in the submitted version of the paper. The language of the manuscript should be improved to understand content in the most part of the manuscript. Provide the z-scale and scanned size of the images in Figure 2.
Author Response
Dear Reviewer,
we kindly ask you to see the attached file, where we placed our responses to your comments.
Kind regarads

Reviewer 2 Report
The authors have investigated insitu drug delivery for treatment of coronary artery disease. Authors have tested the method in two cell lines relevant in atherosclerosis: SMC and EC. The work is relevant for the scientific community. Discussion and interpretation of data corresponds to results obtained.
Minor coments:
Authors should indicate what FITR stands for.
Within paragraph stating at line 186 there are some italic words that should be written in not italic.
The donors who have gave blood, have signed an informed consent?
Legends of figures could be improved. They should be more explanatory
Author Response
Dear Reviewer,
we kindly ask you to see the attached file, where we placed our responses to your comments.
Kind regards

Reviewer 3 Report
The authors synthesized 3 types of silica-based materials via sol-gel technology with different pore size distribution as a matrix for doping with photon and protoporphyrin IX photosensitizers, which can be applied as stent coating for PDT on coronary artery disease treatment. Through various material characterization, toxicity and biological activity assessments, the authors successfully develop efficient material for in situ delivery of photosensitizer for PDT using.
Figure 1a, the authors show that M2 indicates 50% pore size distribution at 3000nm- 8000nm. However, from figure 1a the fitted data is much higher than real data. What model the authors applied to fit data? Can authors clarify why in table 2 the R2still show such high value even the fitted data is far higher than the real one? In figure 7, the authors provide results that photosensitizers were taken up by the cells from M1, M2 and M3 loaded with PPIX and Ph. I would suggest the authors put one more control which is M1, M2 and M3 only with cell lines. At line 464, the authors claim that the lower pore volume suggest better biological efficiency. Can author elaborate more on this? Why it can lead to higher biological activity?
Author Response

(The authors gave the same response as above.)
